# A Comprehensive Molecular and Clinical Analysis of the piRNA Pathway Genes in Ovarian Cancer

**DOI:** 10.3390/cancers13010004

**Published:** 2020-12-22

**Authors:** Eunice Lee, Noor A. Lokman, Martin K. Oehler, Carmela Ricciardelli, Frank Grutzner

**Affiliations:** 1Department of Molecular and Biomedical Sciences, Robinson Research Institute, University of Adelaide, Adelaide, SA 5000, Australia; eunicehsiuyee.lee@adelaide.edu.au; 2Discipline of Obstetrics and Gynaecology, Robinson Research Institute, Adelaide Medical School, University of Adelaide, Adelaide, SA 5000, Australia; noor.lokman@adelaide.edu.au (N.A.L.); martin.oehler@adelaide.edu.au (M.K.O.); 3Future Industries Institute, University of South Australia, Mawson Lakes, SA 5095, Australia; 4Department of Gynaecological Oncology, Royal Adelaide Hospital, Adelaide, SA 5005, Australia

**Keywords:** ovarian cancer, piRNA pathway, patient survival, invasion, follicle stimulating hormone, chemoresistance, therapeutic targets

## Abstract

**Simple Summary:**

Although ovarian cancer (OC) is one of the most lethal gynecological cancers, its development and progression remain poorly understood. The piRNA pathway is important for transposon defense and genome stability. piRNA maturation and function involve a number of genes known as the piRNA pathway genes. These genes have recently been implicated in cancer development and progression but information about their role in OC is limited. Our work aimed to provide a better understanding of the roles of piRNA pathway genes in OC. Through analyzing changes in the abundance of 10 piRNA pathway genes, we discovered gene expression differences in benign vs. cancer, chemosensitive vs. chemoresistant and post hormone treatment in OC samples and cells. Furthermore, we observed the differential effects of these genes on patient survival and OC cell invasion. Overall, this work supports a role of the piRNA pathway genes in OC progression and encourages further study of their clinical relevance.

**Abstract:**

Ovarian cancer (OC) is one of the most lethal gynecological malignancies, yet molecular mechanisms underlying its origin and progression remain poorly understood. With increasing reports of piRNA pathway deregulation in various cancers, we aimed to better understand its role in OC through a comprehensive analysis of key genes: *PIWIL1-4*, *DDX4*, *HENMT1*, *MAEL*, *PLD6*, *TDRD1*,*9* and mutants of *PIWIL1* (*P1∆17*) and *PIWIL2* (*PL2L60*). High-throughput qRT-PCR (*n* = 45) and CSIOVDB (*n* = 3431) showed differential gene expression when comparing benign ovarian tumors, low grade OC and high grade serous OC (HGSOC). Significant correlation of disparate piRNA pathway gene expression levels with better progression free, post-progression free and overall survival suggests a complex role of this pathway in OC. We discovered *PIWIL3* expression in chemosensitive but not chemoresistant primary HGSOC cells, providing a potential target against chemoresistant disease. As a first, we revealed that follicle stimulating hormone increased *PIWIL2* expression in OV-90 cells. *PIWIL1*, *P1∆17*, *PIWIL2*, *PL2L60* and *MAEL* overexpression in vitro and in vivo decreased motility and invasion of OVCAR-3 and OV-90 cells. Interestingly, *P1∆17* and *PL2L60*, induced increased motility and invasion compared to *PIWIL1* and *PIWIL2*. Our results in HGSOC highlight the intricate role piRNA pathway genes play in the development of malignant neoplasms.

## 1. Introduction

Ovarian cancer (OC) is one of the leading causes of death in women worldwide where 184,799 deaths were projected in 2018. This was the second highest mortality among all gynecological cancers [1]. Due to unspecific symptoms and lack of early detection, OC is most commonly diagnosed at an advanced stage and subsequently has very poor prognosis [2,3]. It has been hypothesized that the number of ovulation cycles correlates with the risk of developing OC and that hormones such as follicle stimulating hormone (FSH) and luteinizing hormone (LH) might be involved in the process and induce malignant transformation [4,5]. While incessant ovulation is considered a risk factor, this theory is still controversial [6,7].

OC consists of epithelial ovarian cancers (EOC) (90%) which include subtypes serous, mucinous, clear cell and endometroid. Nonepithelial ovarian cancers (10%) include small-cell carcinomas and sarcomas and malignancies originating from germ or sex cord-stromal cells [8,9]. HGSOC is known for its highly heterogeneous nature, chromosomal instability and high rates of chemoresistance where mutations in the *TP53* gene occur in a majority of HGSOC samples, more than any other cancer [10,11,12,13,14]. While mutations in tumor suppressor *TP53* have been postulated as the driver of HGSOC, the etiology, subsequent progression and development of chemoresistance are poorly understood [15,16]. OC is classified through FIGO staging (stages I–IV) and grading (grades 1–3). In the serous OC context, low grade (grade 1) and high grade (grades 2 and 3) are classified as two separate diseases with different underlying molecular pathology and clinical behavior [11,17,18]. Tumor malignancies are often linked to a large number of genes and their associated mutants as seen in OC [12,19,20]. With the origin and development of OC still under much discussion, the emergence of deregulated piRNA pathway in cancer has led to a few studies investigating their involvement in OC development [6,21,22,23,24,25,26].

The piRNA pathway consists of piRNAs (small noncoding RNAs (ncRNAs)) and a growing list of associated pathway genes which are involved in the biogenesis of piRNAs [27,28,29]. The core biogenesis pathway genes, the *PIWIL* genes (*PIWIL1-4*), are essential in ensuring the maturation of piRNAs through their involvement in either the primary or secondary biogenesis pathway [30]. To do this, PIWILs utilize their three main domains known as PAZ, MID and PIWI. These domains mainly function in binding the 3′ end of piRNAs, 5′ end of piRNAs and having endonuclease activity, respectively [31,32,33]. On top of the *PIWIL* genes, there is a myriad of pathway genes including *DDX4*, *HENMT1*, *MAEL*, *PLD6*, *TDRD1* and *TDRD9*. A more detailed list of pathway genes and their respective functions in piRNA biogenesis was recently reviewed by Ozata, Gainetdinov, Zoch, O’Carroll and Zamore [28]. The piRNA pathway was originally discovered in gonads as a mechanism to control expression of transposable elements (TEs) [34,35]. In addition to having crucial roles in the production of mature piRNAs, the pathway genes can also work in tandem or independently of each other to aid piRNAs in the execution of TE repression [36,37,38,39]. One method is through the fascinating secondary biogenesis pathway where PIWIL2 and PIWIL4 convert TEs into piRNAs, generating a multitude of piRNA species while destroying TEs [36,40,41]. Since the discovery of the piRNA pathway, its roles beyond TE repression has steadily increased especially after reports of its deregulation in cancer [42,43,44].

piRNAs and piRNA pathway genes are increasingly discovered to be involved in various aspects of cancer development and progression [43]. Some of these pathway genes (*PIWIL1*, *PIWIL2*, *TDRD1* and *MAEL*) are categorized as cancer/testis genes due to their restricted expression in testis but in recent years, have been observed to be aberrantly expressed in multiple cancers [43,44,45,46]. The functions of these genes go beyond the piRNA pathway and include regulating cell motility, invasion, proliferation and apoptosis which are hallmarks of cancer progression and malignancy [19,24,43,47,48,49]. Additionally, cancer specific mutations have been revealed including a *PIWIL1* mutant (*P1∆17*) which lacks an exon 17 and a PIWIL2 mutant (PL2L60) with a truncated PAZ domain [25,50]. The *P1∆17* transcript was only present in malignant HGSOC and was proposed to have a premature stop codon with a truncated PIWI domain in its protein form [25]. Upon discovery of PL2L60 in mouse testis, it was then found to be the predominant form of *PIWIL2* in precancerous stem cells where its expression associates with the antiapoptotic *STAT-3*/*BCL-2* pathway [50].

Here, we present extensive expression screening of piRNA pathway genes in early and late stage HGSOC tumors, benign serous cystadenoma as well as low grade OC tumors (workflow diagram, Figure 1). We assessed their potential link to treatment response (chemosensitivity versus chemoresistance), progression-free survival (PFS), post progression-free survival (PPFS) and overall survival (OS) in HGSOC patients. We also determined if there was a change in *PIWIL2* expression after OC cells were treated with FSH and LH, individually and in combination. Finally, we assessed whether overexpression of the pathway genes and mutants (*P1∆17* and *PL2L60*) affected the motility and invasion of HGSOC in vitro and in vivo. This work highlights that the piRNA pathway may function differently in different cancers and provides novel insights into its role in OC.

## 2. Results

### 2.1. Differential Expression of piRNA Pathway Genes Occurs between Benign and Malignant HGSOC Tissue Samples

A growing list of piRNA pathway genes are being implicated in cancer but only limited studies address their potential role in OC. We profiled the mRNA expression levels of 10 piRNA pathway genes in benign serous cystadenoma (*n* = 16), early and late stage HGSOC tissue samples (*n* = 29) (Figure 2). Interestingly, we observed that the *PIWIL* genes had dissimilar trends of expression in benign and cancerous tumors. *PIWIL1* had significantly higher expression (Figure 2a), while *PIWIL2* expression was significantly lower in cancerous tumors (early and late stage) as compared to benign tumors (Figure 2b). Most tumors had no *PIWIL3* expression (Figure 2c) whereas *PIWIL4* had significantly lower expression in early stage HGSOC samples but not late stage when compared to benign tumors (Figure 2d).

RNA helicase *DDX4* had no significant difference in expression when comparing benign to cancerous tumors but there was an increasing trend in expression between early and late stage HGSOC samples (Figure 2e) [51]. RNA methyltransferase *HENMT1* (Figure 2f) and *spindle* class gene *MAEL* (Figure 2g) both had significantly increased expression in HGSOC tumors compared to benign tumors [52,53]. While *HENMT1* expression was significantly increased between benign and early and late stage tumors, *MAEL* expression was only significantly increased between benign and late stage tumors. Phospholipase *PLD6* had similar expression levels in benign and cancerous tumors (Figure 2h) [54]. Methylarginine binding genes *TDRD1* (Figure 2i) and *TDRD9* (Figure 2j), had overall low expression in HGSOC and benign tumors [55]. *TDRD1* expression was reduced in HGSOC compared to benign tumors and expression was significantly lower in late stage tumors compared to benign tumors.

### 2.2. piRNA Pathway Genes are Differentially Expressed in Low Grade and High Grade Ovarian Cancer (All Subtypes)

Following the differential expression observed in HGSOC staging, we next investigated if the piRNA pathway genes had distinct expression levels in low grade (G1) versus high grade (G2, G3) disease in OC (Figure 3). *PIWIL1* and *PIWIL3* had significantly lower expression in high grade compared to G1 (Figure 3a,c). *PIWIL2* and *PIWIL4* expression was not different between low grade and high grade (Figure 3b,d). We also observed similar expression levels for *DDX4*, *MAEL* and *TDRD9* across low and high grade OC (Figure 3e,g,j). In contrast, *HENMT1* expression was significantly higher in G2 and G3 compared to G1 (Figure 3f). Of note, *HENMT1* expression was significantly increased from G2 to G3 (Figure 3f). *PLD6* and *TDRD1* expression were significantly lower in G2 and G3 compared to G1 (Figure 3h,i).

### 2.3. The Level of piRNA Pathway Gene Expression Is Associated with HGSOC Patient Overall and Progression Free Survival

Next, we correlated expression of piRNA pathway genes with early and advanced stage patient outcome using the Kaplan−Meier (KM) online plotter which is based on data from large public microarray databases (Table 1). With the exception of *HENMT1* which did not significantly correlate with PFS, PPFS or OS, all other genes were associated with HGSOC patient outcome. High *PIWIL1* expression was significantly associated with improved PPFS and OS in both early (*p* = 0.034; *p* = 0.017) and late (*p* = 0.0002; *p* = 0.00033) stage HGSOC patients. Conversely, low *PIWIL2* and *PIWIL4* expression were significantly associated with increased PFS in late stage (*p* = 0.0051; *p* = 0.015) patients. HGSOC patients with low *PIWIL2* expression also had significantly increased PPFS (*p* = 0.048).

Although high expression of *DDX4* was associated with longer PFS (*p* = 0.046), low *DDX4* was associated with improved OS (*p* = 0.035) in early stage HGSOC patients. Low *MAEL* expression in both early (*p* = 0.0037) and late (*p* = 0.048) stage HGSOC patients was associated with increased PFS whereas high expression was associated significantly with increased OS (*p* = 0.041) in only late stage HGSOC patients. Low *PLD6* expression was significantly associated with increased OS (*p* = 0.027) in late stage patients. Early stage HGSOC patients with high *TDRD1* expression had very significantly longer PFS (*p* = 0.000042) while late stage patients with low *TDRD9* expression had significantly better PFS (*p* = 0.05).

### 2.4. PIWIL3 Is Expressed in Chemosensitive Primary HGSOC Cells but not in Chemoresistant Primary HGSOC Cells

Recognizing that chemoresistance is a major problem for treatment in OC, we evaluated piRNA pathway gene expression in chemosensitive and chemoresistant HGSOC patient-derived primary cells in search of a potential chemoresistant treatment target [2]. We found that *PIWIL1* and *PIWIL2* had a similar expression pattern in chemosensitive and chemoresistant primary HGSOC cells (Figure 4a,b). Only *PIWIL3* expression was observed to be significantly increased in chemosensitive primary cells (Figure 4c). *PIWIL4*, *DDX4*, *MAEL*, *TDRD1* and *TDRD9* expression decreased in chemoresistant primary cells, but those changes were not significant (Figure 4d,e,g,i,j). *HENMT1* showed no difference in expression while *PLD6* expression had a slight increase in chemoresistant cells compared to chemosensitive cells, albeit without significance (Figure 4f,h).

### 2.5. PIWIL2 Expression Increased upon FSH Treatment in OV-90 Cells

As OC development has been linked to hormones, we were interested to see if the piRNA pathway genes are affected by hormones. RT-PCR of OV-90 cells after treatment with different concentrations of gonadotropins, FSH and LH, showed increased *PIWIL2* expression (Figure 5a). Further analysis using qRT-PCR showed specifically a higher dose of FSH (100 mIU/mL) significantly increased *PIWIL2* expression (Figure 5b). While both treatment doses of LH (25 and 50 mIU/mL) elevated *PIWIL2* expression, this change was not significant (Figure 5b). Interestingly, *PIWIL2* expression after treatment with both LH and FSH (in doses 25 and 50 mIU/mL and 50 and 100 mIU/mL) were lower (not significant) than OV-90 cells treated with 100 mIU/mL FSH alone.

### 2.6. Cells Overexpressing Wildtype and Mutant piRNA Pathway Genes Have Altered Motility and Invasion In Vitro

In several cancer models, knockdown of piRNA pathway genes result in decreased motility and invasion [56,57,58]. Initial work in ovarian cancer indicated an opposite trend where overexpression induced decreased motility and invasion [25]. Here, we assessed the effects of overexpression of *PIWIL1*, *PIWIL2* and *MAEL* as well as known mutants of *PIWIL1* (*P1∆17*) and *PIWIL2* (*PL2L60*) on motility and invasion in vitro. Overexpression of *PIWIL1*, *PIWIL2* and *PL2L60* in OVCAR-3 had significantly decreased motility compared to the empty vector control (Figure 6a). In OV-90, overexpression of the same set of genes showed decreased motility, however, only *PIWIL2* overexpression significantly reduced motility compared to the control (Figure 6b). *P1∆17* overexpression in OVCAR-3 resulted in a more motile phenotype (nonsignificant) while in OV-90 there was no change in cell motility as compared to their respective empty vector control (Figure 6a,b). Comparing the overexpression of wildtype and mutant *PIWIL1* and *PIWIL2* motility in both cell lines, only cells overexpressing *P1∆17* were significantly more motile than *PIWIL1* in OVCAR-3 (Figure 6a). *MAEL* overexpressing OVCAR-3 and OV-90 cells displayed decreased cell motility (nonsignificant) compared to the empty vector control.

Furthermore, OVCAR-3 cells overexpressing *PIWIL1*, *PIWIL2*, *PL2L60* and *MAEL* and OV-90 cells overexpressing *PIWIL2* showed significantly decreased invasion compared to their respective empty vector control (Figure 6c,d). OVCAR-3 cells overexpressing *P1∆17* had increased invasion (nonsignificant) compared to the empty vector control (Figure 6c). Comparison of OVCAR-3 overexpressing wildtype and mutant *PIWIL1* demonstrated that this mutant was more invasive (nonsignificant) (Figure 6c). OV-90 cells overexpressing *PIWIL1*, *P1∆17*, *PL2L60* and *MAEL* were not significantly different in invasiveness compared to empty vector (Figure 6d). While there was no significant difference in invasion between *PIWIL1* and *P1∆17* overexpression in OV-90 cells, overexpression of *PL2L60* was significantly more invasive than *PIWIL2* (Figure 6d).

### 2.7. OV-90 Cells Overexpressing Mutant PIWIL1 Are More Invasive than Wildtype In Vivo

The functional roles of *PIWIL1*, *PIWIL2*, *MAEL*, *P1∆17* and *PL2L60* were further assessed in vivo using the chicken chorioallantoic membrane (CAM) invasion assay. We quantified the number of OV-90 cells that successfully invaded into the ectoderm and beyond as a measure of invasion (Figure 7a–f). We found that *P1∆17* overexpressing OV-90 resulted in a significantly more invasive phenotype as compared to the empty vector control and *PIWIL1* (Figure 7g). *PL2L60*, on the other hand, was more invasive than the empty vector control but not significantly more invasive than *PIWIL2* (Figure 7g). OV-90 cells overexpressing *PIWIL1*, *PIWIL2* and *MAEL* showed a slightly more invasive phenotype (nonsignificant) compared to empty vector in vivo (Figure 7g).

## 3. Discussion

The piRNA pathway is increasingly recognized as an important aspect of cancer development and progression [42,43]. Given the limited information of this pathway in OC, we investigated its expression and effects on OC progression. Expression analysis of 10 selected piRNA pathway genes revealed dynamic changes in relation to OC malignancy, patient survival, and chemoresistance. Additional experiments revealed that FSH treatment on HGSOC cells increased *PIWIL2* expression. Furthermore, overexpression studies on the motility and invasion of HGSOC cells demonstrated that *PIWIL1* or *PIWIL2* mutants can induce a more aggressive phenotype as compared to wildtype in vitro and in vivo.

A number of studies reported that increased expression of piRNA pathway genes contributed to the metastatic progression of various cancers, including OC [56,59,60,61,62]. Limited information is available on OC and while knockdown of pathway genes has been linked to decreased malignancy in vitro, overexpression analysis in OC surprisingly revealed decreased invasion [25,56,57,58]. Our comprehensive analysis revealed a more differentiated pattern where *PIWIL1*, *MAEL* and *HENMT1* had increased expression, while *PIWIL2*, *PIWIL4* and *TDRD1* had decreased expression when comparing benign and malignant tumor samples. This indicates that different aspects of the pathway may be turned up or down as the cancer progresses. For example, PIWIL1 is known to be involved in the primary piRNA biogenesis pathway while PIWIL2 and PIWIL4 participate in the secondary biogenesis pathway. MAEL, HENMT1 and TDRD1 function in both pathways [28,30]. This may be showing that the secondary piRNA biogenesis pathway is specifically downregulated in HGSOC. This also suggests that the pathway genes, having functions beyond piRNA biogenesis, may be behaving independently in malignant tumors. Another possibility is that HGSOC tumors which harbor a high percentage of *TP53* mutations and are very genetically unstable may contribute to why our results contrast other cancers [4,5,6,11,16]. Downregulation of piRNA pathway genes in cancer may be uncommon but have been reported in testicular germ cell tumor (*PIWIL1*, *PIWIL2*, *PIWIL4* and *DDX4*) and renal cell carcinoma (*PIWIL1*, *PIWIL2* and *PIWIL4*) and now, also in our expression study for *PIWIL2*, *PIWIL4* and *TDRD1* [63,64]. Further evidence that the role of this pathway is more complicated comes from piRNAs where the same piRNA species (piRNA-823) can have a suppressive effect in gastric cancer while oncogenic in multiple myeloma [65,66]. Similarly, within gastric cancer, it was observed that different piRNA species can react to support (piRNA-651) and inhibit (piRNA-823) cancer cell growth [65,67]. The rapidly increasing body of work highlights the importance of ncRNAs in terms of oncogenicity or tumor suppression in various cancers including OC. For example, different species of microRNAs (miRNAs) can also have oncogenic or tumor suppressive roles in OC [68]. This can be linked to the piRNA pathway and miRNAs regulating transcription factors ZEB1,2 and SNAI1-3 which affect cell epithelial to mesenchymal transition (EMT), enabling the promotion or repression of cell invasion and metastasis [47,68,69]. Such piRNA and miRNA species involved in the molecular regulation of cancer progression could be potential targets for OC treatment. Hence, if piRNAs, as part of the piRNA pathway, could function to promote and suppress cancer, it is possible that the pathway genes act in a similar fashion. Together, our findings add to the mounting evidence that the piRNA pathway can act in different ways in different cancers or even different diseases of the same cancer as we demonstrated in OC.

It is accepted that low grade and high grade OC are considered different diseases. Interestingly, this is also reflected in the piRNA pathway expression profile demonstrated by the significantly lower expression of *PIWIL1*, *PIWIL3*, *PLD6* and *TDRD1* in high grade as compared to low grade. *HENMT1*, on the other hand, had elevated expression in high grade compared to low grade OC. The molecular action for PIWIL3 and PLD6 in low grade OC could be involvement in growth regulation such as in glioma and mediating downstream pathways for proto-oncogene, MYC, as seen in breast cancer, respectively [70,71]. At present, there are no studies of *TDRD1* in OC however, *TDRD1* is now considered a potential biomarker for prostate cancer as it strongly associates with expression of the frequently mutated transcription factor, *ERG* [23,49]. Thus, *TDRD1* may also be a biomarker for low grade OC. Implications of aberrant *HENMT1* expression in cancer are unknown but it has known roles in maintaining piRNA stability by 2′-O-methylation of piRNAs and ensuring TE repression [39,72]. This analysis suggests that piRNA pathway genes may have expression profiles specific to the histotype of OC. Interestingly, a microarray study demonstrated similar OC histotype-specific miRNA signatures, supporting the diversity of ncRNA expression in different OC histotypes and highlighting the need for further studies [73].

Next, we assessed whether piRNA pathway gene expression levels were associated with HGSOC progression and patient survival using the KM plotter. High *PIWIL1* expression was linked to better PPFS and OS. In contrast, patients with low *PIWIL2* and *PIWIL4* expression had better PFS. With limited patient data in our gene expression study, we were unable to compare if the lower expressing *PIWIL2* and *PIWIL4* samples remained progression free as seen using the KM online plotter. However, the difference in expression level for *PIWIL1* versus *PIWIL2* and *PIWIL4* contributing to better patient outcome was expected given the difference we observed when comparing their expression in benign and malignant tumors as well as low grade OC versus high grade OC. This supports the possibility that the *PIWIL* genes could be behaving differently in HGSOC. *DDX4* and *MAEL* had conflicting results as they both had low and high expression significantly linked to better prognosis of PFS and OS. The current literature on *DDX4* suggests that it plays a role in OC progression due to its influence on DNA damage checkpoints and association with cancer stem cell marker, CD133 [59,60]. Our results showed higher *MAEL* expression with increasing tumor stage. The association of low *MAEL* expression with longer PFS in early and late stage HGSOC patients may be related to its putative role in other cancers of promoting EMT and preventing apoptosis and DNA damage in cancer cells [22,47]. However, we observed that high *MAEL* expression was associated with increased OS in late stage HGSOC. This is consistent with our previous and current work where *MAEL* overexpression in EOC decreased invasion [25]. High *TDRD1* expression was strongly correlated with better PFS but it is currently unknown what its functional role is in cancer other than strong association with *ERG* expression in prostate cancer [49,74]. On the other hand, low *TDRD9* expression was associated with longer PFS. This is consistent with a study where *TDRD9* knockdown impaired proliferation of two lung cancer cell lines [61].

The prevalence of chemoresistance in OC and late diagnosis have contributed to its concerningly low 5-year survival rates (26–42%) and label as one of the most lethal gynecologic malignancies [2]. The mechanisms leading to the development of resistance has yet to be elucidated. Here, we discovered that *PIWIL3* was not expressed in chemoresistant HGSOC cells but, interestingly, was expressed in chemosensitive cells. This finding agrees with previous work demonstrating lower *PIWIL3* expression in benign and malignant EOC as compared to in a normal ovary which points towards reduced *PIWIL3* expression possibly being involved in the progression of OC [25]. In contrast, *PIWIL3* had increased expression in more malignant tumors of gastric cancer and melanoma wherein overexpression leads to proliferation and invasion [56,75]. Further conflicting results showed *PIWIL3* overexpression induced glioma regression whereas knockdown of *PIWIL3* in pancreatic cancer conferred sensitivity to resistant cells [70,76]. Here, we reason that *PIWIL3* may be silenced in HGSOC cells as it may play a role in genomic stability through its putative role in piRNA biogenesis [77]. Extensive genomic instability can confer sensitivity to chemotherapy [78,79]. Additional work will be needed to address its role in sensitizing HGSOC cells and the mechanism of action in the piRNA pathway. Keeping in mind that our sample size was small, it may be worthwhile exploring *PIWIL4*, *MAEL*, *PLD6*, *TDRD1* and *TDRD9* as potential targets in addition to *PIWIL3* for overcoming chemotherapy resistance.

The effects of hormones on the piRNA pathway have been reported in mice, frogs and fish with only one study in breast and prostate cancer [80,81,82,83,84]. Exploring this novel avenue in OC, we exposed a HGSOC cell line, OV-90, with varying concentrations of FSH and LH. We found that a higher concentration of FSH was able to stimulate increased *PIWIL2* expression but none of the treatments stimulated a change in *PIWIL1* or *MAEL* expression [85] *PIWIL2* had no change in expression when treated with a high dose of combined FSH and LH. This could be explained by their antagonistic nature where FSH induced cell proliferation was blocked when LH was introduced in EOC cell lines [86]. We postulate that PIWIL2 may be a downstream target of FSH as they both affect proteins involved in cell proliferation and EMT pathways namely, STAT3 and MMP-9 [57,87,88]. Another set of proteins that PIWIL2 and FSH are involved in regulating are stem cell markers, OCT4, NANOG and SOX-2 [89,90,91]. In addition, induction of *PIWIL2* was also reported to endow cancer stem cell-like properties to human fibroblasts with ectopic expression of these stemness markers [90]. While there are many papers on FSH causing OC proliferation and migration [4,92,93,94,95], there is a study claiming FSH as having protective properties against cancer [96]. Therefore, further investigation is needed to identify the relationship between FSH and PIWIL2 and whether they contribute to or against the development of HGSOC.

To establish the effects of increased piRNA pathway gene expression on HGSOC motility and invasion in vitro and in vivo, we assessed this in 2 HGSOC cell lines, OVCAR-3 and OV-90, both harboring a *TP53* missense mutation [97]. Our previous work on SKOV-3 overexpressing *PIWIL1* and *MAEL* showed a decrease in invasion in vitro [25]. Here, we expanded the overexpression study to include *PIWIL2*, *P1∆17* and *PL2L60* in cell lines, OVCAR-3 and OV-90. As well as demonstrating similar results for *PIWIL1* and *MAEL* as SKOV-3 cells, we revealed that OVCAR-3 and OV-90 cells overexpressing *PIWIL2*, *P1∆17* and *PL2L60* also had decreased motility and invasion compared to empty vector transfected cells. This is unexpected as in most cancers, the overexpression of these wildtype and mutant piRNA pathway genes were associated with a more invasive phenotype [50,98,99,100]. Key cancer-related pathways and proteins that these genes can regulate include the AKT/GSK-3*β*/SNAIL pathway, MMP-9, CD44 and STMN1 which are involved in EMT and metastasis [47,57,88,89,99]. There is a possibility that these pathways and proteins react differently in HGSOC due to genomic instability and the presence of TP53 mutants that could either have loss-of-function, gain-of-function or dominant negative mutations [16]. More importantly, TP53 mutants are more resistant to degradation than the wildtype form, hence could cause different outcomes regardless of the presence of wildtype TP53 [16]. There is evidence that PIWIL2 can form a complex with STAT3 and c-SRC through interaction with its PAZ domain to repress *TP53* transcription [101,102]. Could PIWIL2 have a protective role by downregulating mutant *TP53* in HGSOC? It is interesting that OVCAR-3 and OV-90 cells overexpressing *P1∆17* and *PL2L60* were more invasive as compared to cells overexpressing *PIWIL1* and *PIWIL2*. This may be attributed to the truncation of their PIWI and PAZ domains, respectively [25,50]. It is possible that truncation of these domains could lead to impaired piRNA binding, endonuclease activity and interaction with the growing list of oncogenes and tumor suppressors. 

## 4. Materials and Methods

### 4.1. Patient Cohort

Patient tissue samples were obtained with written consent and approval by the Central Adelaide Local Health Network Human Research Ethics Committee (RAH Protocol #140201). Clinicopathological characteristics of the patient samples used in the Fluidigm qRT-PCR expression analysis are listed in Appendix A. Ethics Committee approval was obtained on 13 January 2014. Formalin-fixed paraffin-embedded (FFPE) tissue sections of patients were stained with hematoxylin and eosin to confirm cancer content [85] RNA extractions were then carried out on frozen patient tissues which FFPE sections had at least 50% cancer content.

### 4.2. RNA Extraction, cDNA Synthesis and Quantification

RNA extractions and cDNA syntheses were carried out on all samples using TRIzol (Thermo Fisher Scientific, Waltham, MA, USA) and iScript cDNA synthesis kit (Bio-Rad Laboratories, Hercules, CA, USA) according to the manufacturers’ protocol. RNA samples were quantified using Qubit (Invitrogen, Waltham, MA, USA) where 400 ng was cDNA synthesized and 2 µL of cDNA was used in the Fluidigm high throughput qRT-PCR. For the RT-PCRs and other qRT-PCRs, RNA samples were quantified using NanoDrop (Thermo Fisher Scientific, Waltham, MA, USA). A 1 µg sample of RNA was used for each cDNA synthesis reaction and diluted 1 in 5 after the reaction. A sample of 5 µL was then used in the RT-PCRs and 2 µL in the other qRT-PCRs.

### 4.3. Primary and Established Ovarian Cancer Cell Lines

Chemosensitive (*n* = 5) and chemoresistant (*n* = 7) primary cells were obtained from ascites of high grade OC patients before and after chemotherapy treatment as previously described [103]. Chemoresistance was determined when patients relapse and no longer respond to chemotherapy. Chemosensitivity was classified as patients responding to chemotherapy and not progressing within 6 months of completing their treatment. These primary cells were grown in Advanced RPMI 1640 media (Gibco, Waltham, MA, USA, cat no. 12633-020) containing 10% fetal calf serum (FCS) (Scientifix, Clayton, VIC, Australia) and 1% each of penicillin/streptomycin (Life Technologies, Mulgrave, VIC, Australia), fungizone (Sigma−Aldrich, St. Louis, MO, USA) and glutamax (Life Technologies, Mulgrave, VIC, Australia) and maintained at 37 °C in a 6% CO_2_ environment.

Human HGSOC cell lines, OVCAR-3 and OV-90, were purchased from the American Type Culture Collection (ATCC, Manassas, VA, USA). They were cultured in RPMI 1640 (Gibco, Waltham, MA, USA, cat no. 11875-093) with FCS (5% for OVCAR-3, 10% for OV-90), 1% penicillin/streptomycin and 1% glutamax.

### 4.4. Quantitative Real-Time PCR (qRT-PCR)

To assess piRNA pathway gene expression in HGSOC tumors, chemosensitive and chemoresistant primary HGSOC cells, a high throughput gene expression qRT-PCR (Fluidigm, South San Francisco, CA, USA) was performed at the Australian Cancer Research Foundation Cancer Genomics Facility using the 96.96 Dynamic Array integrated fluidic circuit (IFC) (Integrated Sciences, Chatswood, NSW, Australia). The 20X Taqman assays (listed in Appendix A), 2 µL each, were pooled and cDNA preamplified under cycling conditions: 95 °C for 2 min then 14 cycles of 95 °C for 15 s lastly, 60 °C for 4 min on the C1000 cycler (Bio-Rad Laboratories, Hercules, CA, USA). IFC Controller MX (Fluidigm, South San Francisco, CA, USA) was used to prime the IFC with control line fluid (Fluidigm, South San Francisco, CA, USA) using the Prime (136×) script. Load Mix (136×) script was utilized to load samples and assays. A premix containing preamplified cDNA, 2X Quanta PerfeCTa qPCR Fast Mix, low ROX (Quanta BioSciences, Beverly, MA, USA) and 20X GE Sample Loading Reagent (Fluidigm, South San Francisco, CA, USA) was made. The premix and Taqman assays with 2X Assay Loading Reagent (Fluidigm, South San Francisco, CA, USA) were loaded into individual inlets. Thermal cycling then started in the BioMark HD System (Fluidigm, South San Francisco, CA, USA) with 45 °C for 2 min, thermal mix: 70 °C for 40 min and 60 °C for 30 s, hot start: 98 °C for 1 min, followed by 35 cycles of 97 °C for 5 s and lastly, 60 °C for 20 s. Gene expression data was collected using the BioMark HD Data Collection software (Fluidigm, South San Francisco, CA, USA). Visualization and exportation of data were performed using the Fluidigm Real-Time PCR Analysis software (Fluidigm, South San Francisco, CA, USA). The 2^−∆CT^ method was used to normalize the Ct values to the geomean of the Ct of housekeepers, *TBP* and *GUSB*.

To confirm *PIWIL2* expression after hormone treatment on OV-90 cells, qRT-PCR was carried out using Taqman assays *HPRT1* and *PIWIL2* (Appendix A). Technical duplicates of Taqman gene expression master mix, respective Taqman assays and cDNA were loaded in the MicroAmp Fast Optical 96-well plate (Applied Biosystems, Waltham, MA, USA) for thermal cycling in the StepOnePlus System (Applied Biosystems, Waltham, MA, USA). Cycling conditions were: 50 °C for 2 min, 95 °C for 10 min, 40 cycles of 95 °C for 15 s and 60 °C for 1 min. The 2^−∆∆CT^ method was used to normalize *PIWIL2* Ct values to housekeeper, *HPRT1*.

GraphPad Prism 8 was used for all statistical analysis. Normality of data was tested using the Shapiro−Wilk test followed by either Kruskal−Wallis (non-normal distribution) or one-way ANOVA (normal distribution) for comparing expression in benign, early and late stage HGSOC samples and Mann−Whitney U (non-normal distribution) or unpaired *t*-test (normal distribution) for comparing expression of chemosenstive and chemoresistant primary cells in the Fluidigm qRT-PCR. Mann−Whitney U test (non-normal distribution) was used to analyze the expression differences between benign and cancerous samples. Data from the other qRT-PCRs were subjected to the Shapiro−Wilk test followed by unpaired *t*-test as the data was normally distributed. 

### 4.5. Public Microarray Online Databases

Expression profiles of the piRNA pathway genes across OC grades 1–3 were evaluated using CSIOVDB (http://csiovdb.mc.ntu.edu.tw/CSIOVDB.html), a microarray gene expression database *n* = 3431 [104]. The OC subtypes analyzed consisted of clear cell, endometrioid, mucinous, and serous. The Kaplan−Meier plotter (http://kmplot.com/analysis/) was used to examine the relationship with PFS, PPFS and OS for OC (*n* = 2190) patients and expression levels of the piRNA pathway genes (Appendix A) [105]. There was no microarray data available for *PIWIL3*. Parameters were set in accordance with the Fluidigm samples of HGSOC (grades 2 and 3). Early stage consisted of stage 1 and 2 while the late stage consisted of only stage 3 HGSOC patients. Patients were split by the best cut-off selected by the online plotter tool.

### 4.6. Overexpression Transfections

Transfections on OVCAR-3 and OV-90 were carried out using Attractene (Qiagen, Chadstone, VIC, Australia) according to the manufacturer’s protocol. To ensure better survival and transfection efficiency, the cells were seeded 48 h pre-transfection at 40–50% confluency and the culture media was replaced with Opti-MEM (Gibco, Waltham, MA, USA, cat no. 11835-030) for DNA and Attractene complex formation. The passage number was kept below 10 post-thaw for all transfections to reduce variability in experiments. Cells were harvested 60–65 h post transfection for subsequent assays. pcDNA3.1 (+) plasmid (Invitrogen, Waltham, MA, USA) was used as the empty vector control. Overexpression constructs were *PIWIL1* (oHu24048), *PIWIL2* (oHu26193) and *MAEL* (oHu11219) inserted in pcDNA3.1 (+) plasmids (GenScript, Piscataway, New Jersey, USA), respectively. Overexpression mutant constructs, *P1∆17* and *PL2L60* (GenScript, Piscataway, New Jersey, USA), were made by removing exon 17 from *PIWIL1* cDNA (ENST00000245255.7) and a readily available *PL2L60* cDNA sequence (AK027497.1) inserted in pcDNA3.1 (+) plasmids.

### 4.7. RT-PCR

RT-PCRs were performed on a 94 °C preheated C1000 Touch thermal cycler (Bio-Rad Laboratories, Hercules, CA, USA) with cycling conditions: 94 °C for 30 s, 35 cycles of 94 °C for 30 s, 55 °C (β*-actin*)/59 °C (*PIWIL2*) for 30 s, 68 °C for 1 min followed by a final extension of 68 °C for 5 min. The composition of each 25 µL reaction using One*Taq* DNA polymerase (New England Biolabs, Ipswich, MA, USA) were following manufacturer’s protocol. The ChemiDoc MP Imager (Bio-Rad Laboratories, Hercules, CA, USA) was used to visualize PCR products on 2% agarose gels stained with ethidium bromide and generate images. β*-actin* and *PIWIL2* primers (Appendix A) were purchased from Integrated DNA Technologies.

### 4.8. In Vitro Motility and Invasion Assay

The ChemoTx^®^ 96-well plate (Neuroprobe, Gaithersburg, MD, USA) was used to assess motility of transfected OVCAR-3 and OV-90 cells. Addition of an even spread of dried 0.6 µL Geltrex (Gibco, Waltham, MA, USA) diluted 1:1 with media (RPMI1640 + 0.1% BSA) on the filter membrane was used to determine invasion. Briefly, cells were labelled with calcein AM (Life Technologies, Mulgrave, VIC, Australia) after 30 min of incubation in the dark on a nutator. Excess calcein AM were removed by washing the cells twice with media (RPMI1640 + 0.1% BSA). Portions of 4 × 10^4^ cells were then pipetted onto each pore of the filter above a microplate containing wells prefilled with chemoattractant (10% FCS) and media (RPMI1640 + 0.1% BSA). Reverse pipetting was employed at every step to prevent bubble entrapment. After a 6-hour, 37 °C incubation, cells that had migrated or invaded the filter were measured using the Triad series multimode detector (Dynex Technologies, Chantilly, VA, USA) at 485–520 nm. Assays were carried out in biological triplicate and had technical replicates to a total of *n* = 21–24 per cell line per construct transfected. Statistical analysis was carried out on GraphPad Prism 8 first by Shapiro−Wilk test of normality followed by either Mann−Whitney U test (non-normal distribution) or unpaired *t*-test (normal distribution).

### 4.9. Chicken Chorioallantoic Membrane (CAM) Assay and Immunohistochemistry

The CAM assays for OV-90 cells were carried out as described previously [106]. CAM harvested from each embryo had 1–2 separate onplants consisting of a mixture of 90,000 cells and matrigel (*n* = 4–18 per construct transfected) for analysis. Paraffin serial sections (5 µm) were stained with hematoxylin and eosin to determine the best paired section for immunohistochemistry (IHC). Paraffin sections for IHC were firstly placed on a 60 °C heat plate for a minimum of 1 h then dewaxed with 100% xylene and ethanol and washed in 1X PBS. After incubating for 5 min in 1% H_2_O_2_ and 1X PBS washes, citrate buffer antigen retrieval was performed in a steam microwave (Sixth Sense, Whirlpool, Dandenong South, VIC, Australia) for 10 min. Sections were incubated for 30 min in blocking buffer (5% goat serum in 1X PBS) before an overnight incubation with 1:800 CD44 antibody (Thermo Fisher Scientific, Waltham, MA, USA, cat no. #MA5-13890) in blocking buffer. With 1X PBS washes in between, sections underwent an hour incubation of 1:400 goat anti-mouse antibody (Dako, Agilent, Santa Clara, CA, USA, cat no. #E0433) in blocking buffer, an hour incubation of 1:500 streptavidin (Dako, Agilent, Santa Clara, CA, USA, cat no. #P0397) in 1X PBS and 6 min of 1:1 DAB and H_2_O_2_ mixture. Lastly, sections were counterstained with hematoxylin, dipped in 70% ethanol, 100% ethanol, 100% xylene and mounted with Pertex mounting medium (HD Scientific, Ringwood, VIC, Australia). After drying, slides were scanned with the Nanozoomer Digital Pathology System (Hamamatsu Photonics, Hamamatsu City, Shizuoka Prefecture, Japan). The NDP scan software v2.2 (Hamamatsu Photonics, Hamamatsu City, Shizuoka Prefecture, Japan) was used to collect slide images and quantify cells that crossed the ectoderm as a measure of invasion. CAM assays were carried out on biological duplicates. Statistical analysis was carried out on GraphPad Prism 8 first by Shapiro−Wilk test of normality followed by either Mann−Whitney U test (non-normal distribution) or unpaired *t*-test (normal distribution). Ethical approval was obtained on 5 October 2018 from the University of Adelaide Animal Ethics Committee (#33109).

### 4.10. FSH and LH Treatment

OV-90 cells (1 × 10^6^) were seeded in T25 flasks and cultured for at least 24 h in normal culture media. Next, the flasks were rinsed with phenol red-free RPMI 1640 (Gibco, Waltham, MA, USA, cat no. 11835-030) and cultured in “complete” media for a minimum of 24 h. “Complete” media consisted of phenol red-free RPMI 1640, 1% each of AlbuMAX II 20% solution (Life Technologies, Mulgrave, VIC, Australia, cat no. E003000PJ), SPITE 100X (Sigma−Aldrich, St. Louis, MO, USA, cat no. S5666), penicillin/streptomycin, glutamax and 0.2% fungizone. Hormone treatment media then replaced the “complete” media for 24 h before RNA was harvested for subsequent RT-PCR and qRT-PCR. The untreated control was maintained in “complete” media. Hormone treatments were made up in “complete” media: 50 mIU/mL FSH, 100 mIU/mL FSH, 25 mIU/mL LH, 50 mIU/mL LH, 50 mIU/mL FSH + 25 mIU/mL LH and 100 mIU/mL FSH + 50 mIU/mL LH. FSH (AFP-7298A) and LH (AFP-4395A) were purchased from the National Hormone and Peptide Program (Torrance, CA, USA).

## 5. Conclusions

In summary, our extensive analysis of piRNA pathway genes in OC adds to the growing knowledge base about the role of this pathway in cancer. Differential expression of these piRNA pathway genes in OC depicts a more complex association of their expression with various aspects of malignancy, patient survival, chemoresistance, gonadotropin treatment and invasion. The observation that piRNA pathway genes may act differently in the HGSOC context compared to other cancers is possibly due to its exceptionally high proportion of *TP53* mutations, genomic instability and highly heterogeneous nature. Here, we present an expression profile of 10 piRNA pathway genes in benign ovarian tumors, low grade OC and HGSOC and their effects on patient survival. We identified *PIWIL3* as a potential target for chemoresistance in HGSOC. Further, we made a novel discovery that FSH can mediate *PIWIL2* expression. Together with in vitro and in vivo studies, we revealed that overexpression of *PIWIL1*, *P1∆17*, *PIWIL2*, *PL2L60* and *MAEL* caused a decrease in motility and invasion. However, mutants *P1∆17* and *PL2L60* surprisingly induced a more invasive phenotype compared to wildtype. While providing a deeper understanding, this work also raises a number of questions about the complicated relationship between piRNAs, the pathway genes and their functions in OC pathogenesis. Additional work is required to further explore the clinical relevance of the variability in piRNAs and pathway genes in OC patients.

## Figures and Tables

**Figure 1 cancers-13-00004-f001:**
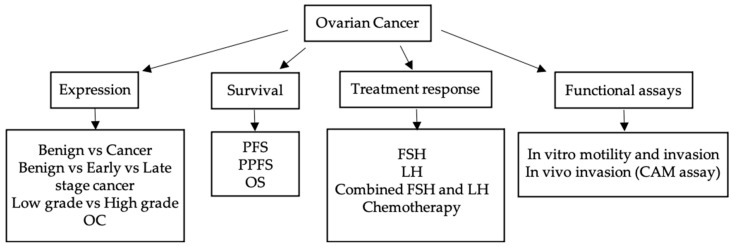
Workflow diagram of this study showing analysis performed for investigating expression differences and effects of the piRNA pathway genes in multiple aspects of OC.

**Figure 2 cancers-13-00004-f002:**
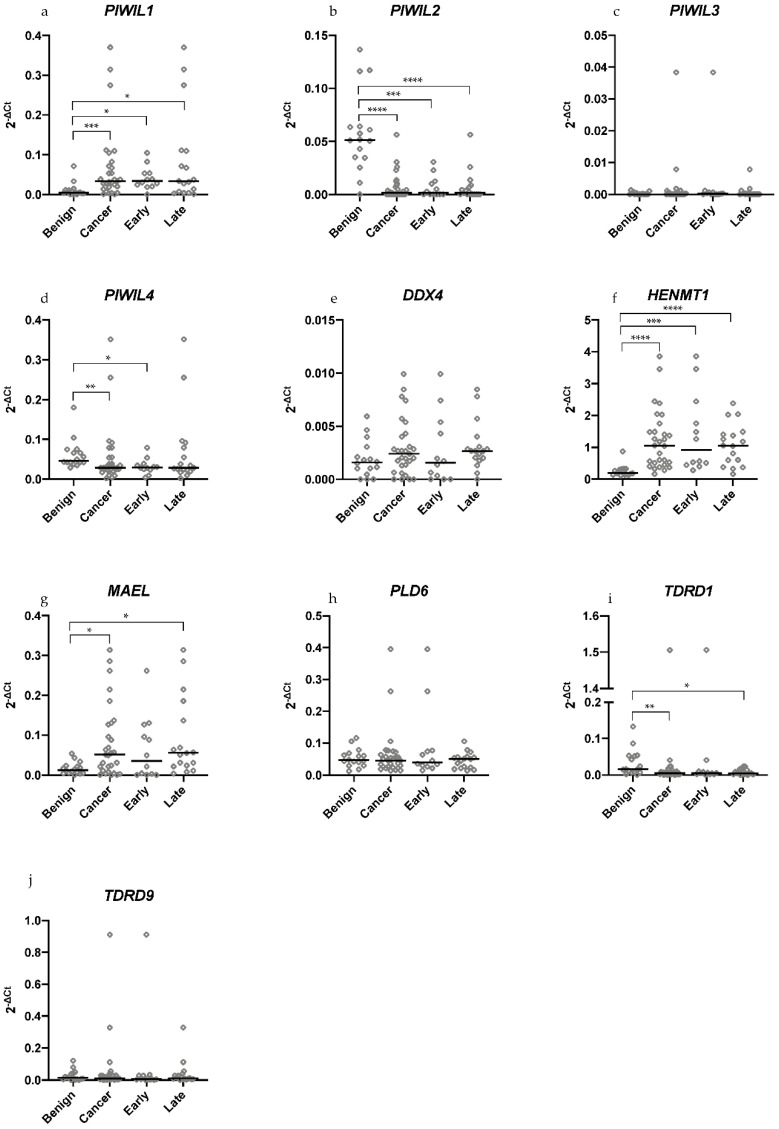
piRNA pathway gene expression in benign serous cystadenoma and high grade serous ovarian cancer (HGSOC) tumors. (**a**–**d**) Among the 4 *PIWIL* genes, only *PIWIL3* had no significant difference between benign and cancerous tumors. *PIWIL1* had increased expression while *PIWIL2* and *PIWIL4* had decreased expression between benign and cancerous samples. (**e**–**j**) Comparing expression of benign to cancerous tumors, *HENMT1* and *MAEL* had significantly increased expression while *TDRD1* expression was significantly decreased. *DDX4*, *PLD6* and *TDRD9* presented similar expression levels across benign and cancerous tumors. All Ct scores were normalized to the geomean of housekeepers, *TBP* and *GUSB*. A median line was used to plot the respective dataset. Benign (*n* = 16), cancer (early and late HGSOC, *n* = 29), early HGSOC (stages 1,2, *n* = 12) and late HGSOC (stage 3, *n* = 17). Statistical tests Mann−Whitney U was used to analyze benign versus cancer while Kruskal−Wallis (non-normal distribution) or one-way ANOVA (normal distribution) were used accordingly when comparing expression differences in benign, early and late stage HGSOC. * *p* < 0.05; ** *p* < 0.005; *** *p* < 0.0005; **** *p* < 0.0001.

**Figure 3 cancers-13-00004-f003:**
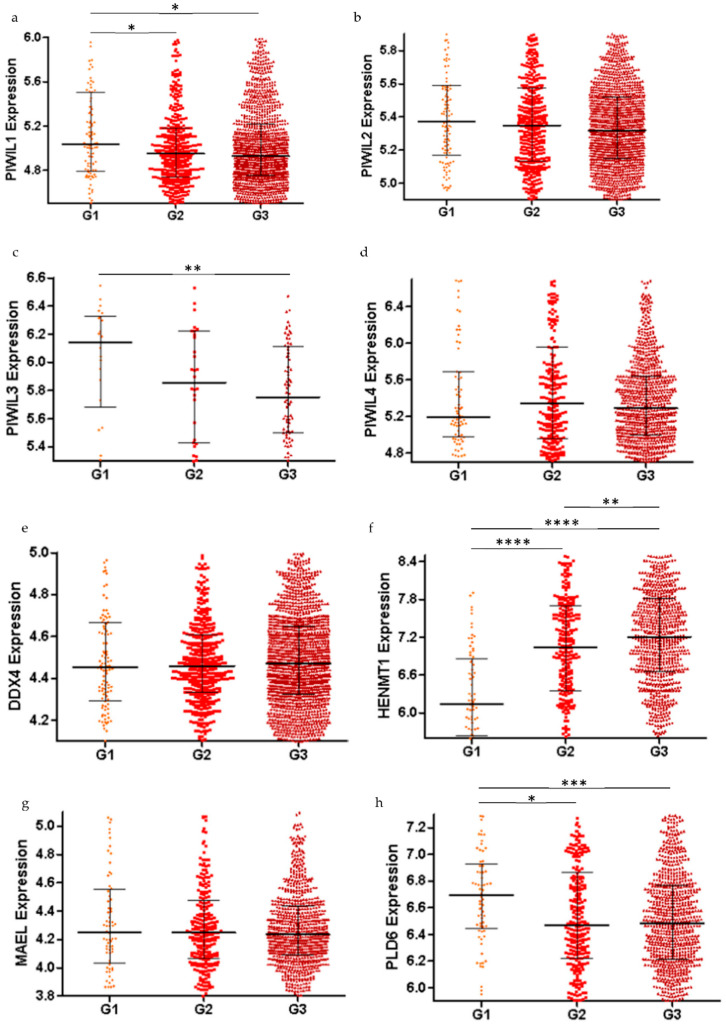
Using the CSIOVDB public microarray database (*n* = 3431) to determine piRNA pathway gene expression in grades 1, 2 and 3 (G1, G2 and G3) of all ovarian cancer subtypes. (**a**–**d**) Among the 4 *PIWIL* genes, *PIWIL1* and *PIWIL3* showed significantly lower expression in the high grade (G2/G3) disease as compared to low grade (G1). *PIWIL2* and *PIWIL4* had similar expression levels in G1, G2 and G3 ovarian cancer samples. (**e**,**g**–**j**) *PLD6* and *TDRD1* also had significantly decreased expression when comparing low grade to high grade. *DDX4*, *MAEL* and *TDRD9* had no significant difference in expression across G1, G2 and G3. (**f**) *HENMT1*, in contrast, had significantly higher expression in G2 and G3 as compared to G1. Mann−Whitney U test was used to assess significance. * *p* < 0.05; ** *p* < 0.005; *** *p* < 0.0005; **** *p* < 0.0001.

**Figure 4 cancers-13-00004-f004:**
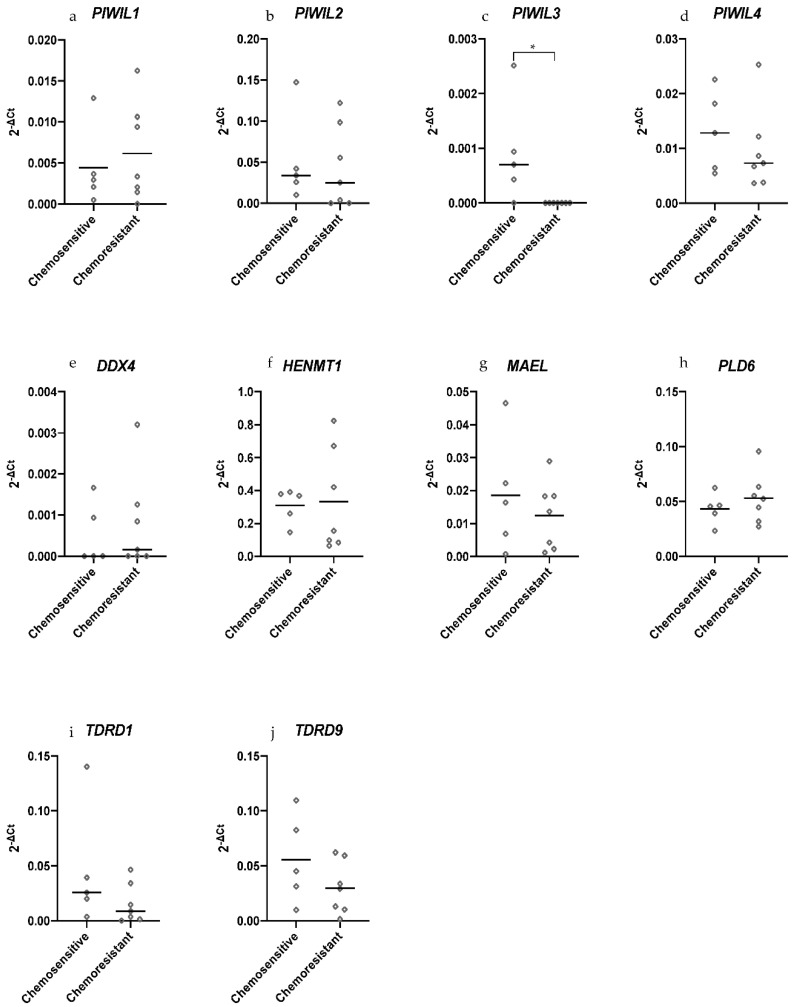
piRNA pathway gene expression in chemosensitive (*n* = 5) and chemoresistant (*n* = 7) primary HGSOC cells. (**c**) *PIWIL3* was not expressed in chemoresistant cells but was expressed in most of the chemosensitive cells (*n* = 4). (**a**,**b**,**d**–**j**) While the other genes had no significant change in expression, it was observed that they had varying degrees of differential expression. All Ct scores were normalized to the geomean of housekeepers, *TBP* and *GUSB*. Either a median (non-normal distribution) or mean (normal distribution) line was plotted. Statistical tests Mann−Whitney U (non-normal distribution) or unpaired *t*-test (normal distribution) were used accordingly. * *p* < 0.05.

**Figure 5 cancers-13-00004-f005:**
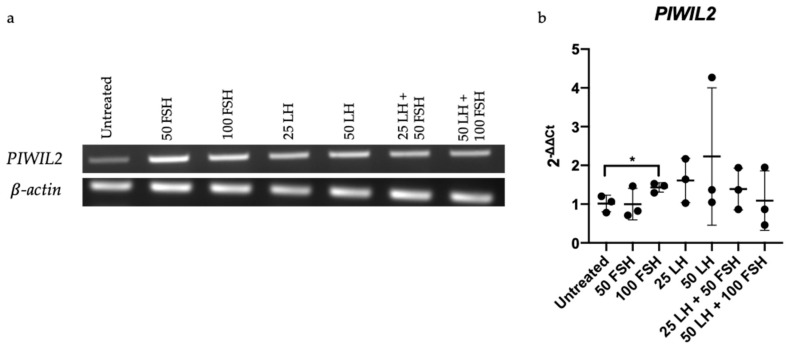
*PIWIL2* expression in OV-90 cells before and after a 24-h follicle stimulating hormone (FSH) and luteinizing hormone (LH) treatment. (**a**) RT-PCR: *PIWIL2* expression increased upon individual and combination treatment of FSH and LH. (**b**) qRT-PCR: *PIWIL2* had a significant increase in expression after 100 mIU/mL FSH treatment. Ct scores were normalized to housekeeper, *HPRT1* and the average of untreated cells for 2^−∆∆CT^. Each colored circle represents one biological replicate with technical duplicate where three biological replicates were carried out per treatment group. Labels with numerical values indicate concentration (mIU/mL) of hormone used. Mean with standard deviation was plotted. Unpaired *t*-test was used to assess significance. * *p* < 0.05.

**Figure 6 cancers-13-00004-f006:**
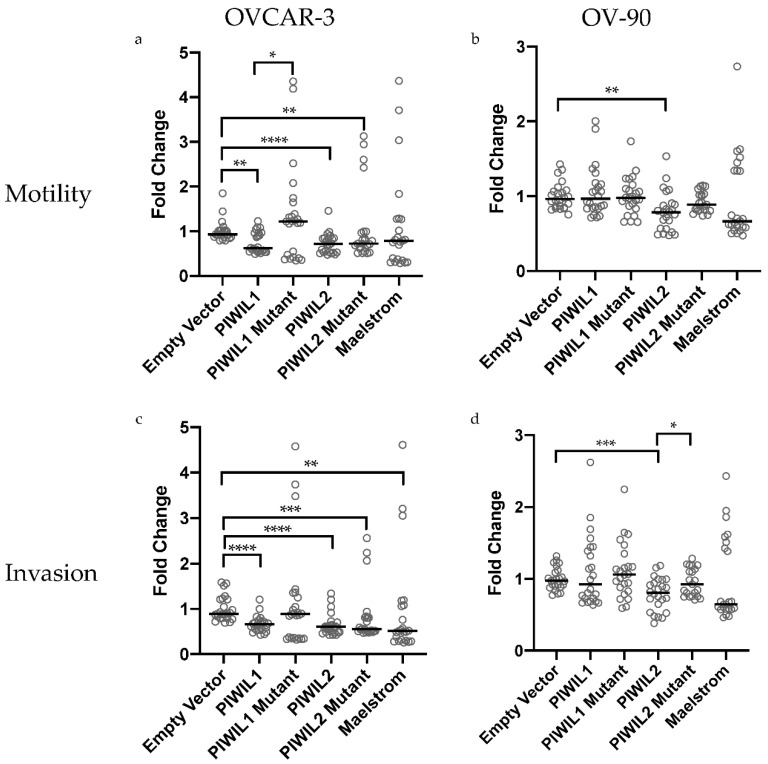
In vitro motility and invasion of OVCAR-3 and OV-90 cells after overexpressing piRNA pathway genes and its mutants (*n* = 21–24 per cell line per transfection). (**a**,**b**) Motility: *PIWIL1* and *PL2L60* (*PIWIL2* mutant) overexpression significantly decreased OVCAR-3 motility while *PIWIL2* overexpression significantly decreased motility in both OVCAR-3 and OV-90. OVCAR-3 cells overexpressing *P1∆17* (*PIWIL1* mutant) had increased motility compared to empty vector and was significantly more motile than *PIWIL1* overexpression. (**c**,**d**) Invasion: overexpression of *PIWIL1*, *PIWIL2*, *PL2L60* and *MAEL* significantly decreased OVCAR-3 invasion while *PIWIL2* overexpression significantly decreased OV-90 invasion. OV-90 cells overexpressing *PL2L60* had significantly increased invasion compared to *PIWIL2* overexpression. All fold changes were calculated relative to empty vector. Three biological replicates were carried out per cell line per transfection group. Bar represents median values plotted for all the datasets. Statistical tests Mann−Whitney U (non-normal distribution) or unpaired *t*-test (normal distribution) were used accordingly. * *p* < 0.05; ** *p* < 0.005; *** *p* < 0.0005; **** *p* < 0.0001.

**Figure 7 cancers-13-00004-f007:**
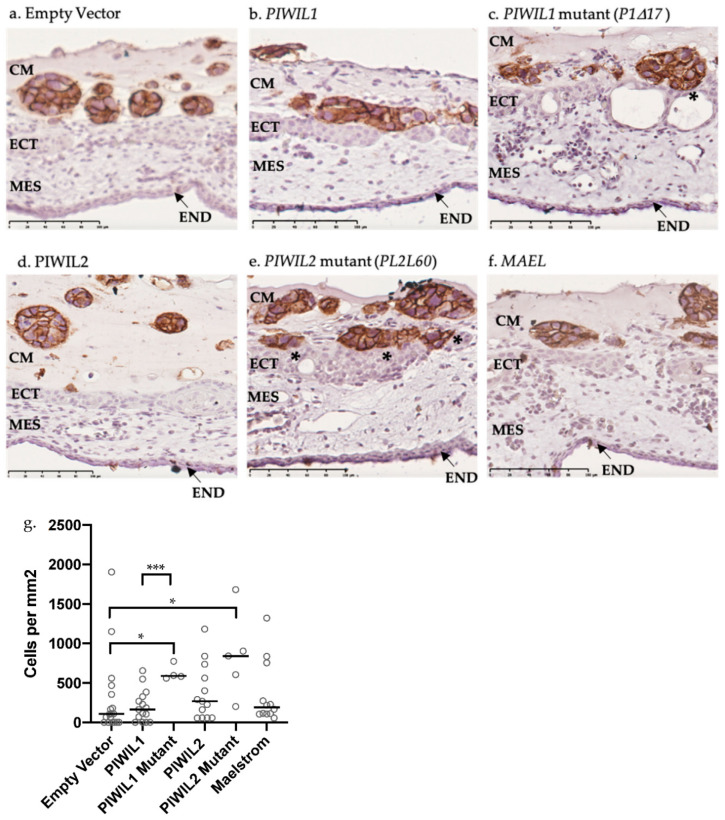
The in vivo invasion analysis using the chicken chorioallantoic membrane (CAM) assay. (**a**–**f**) CD44 stained OV-90 cells transfected with (**a**) empty vector (pcDNA3.1(+)) and overexpression constructs, (**b**) *PIWIL1*, (**c**) *P1∆17*, (**d**) *PIWIL2*, (**e**) *PL2L60* and (**f**) *MAEL*. CAMs were counterstained with haematoxylin. Invasion was quantified by the number of OV-90 cells that successfully breached the ECT and beyond (* marked next to breached area). CM = cancer cells mixed with matrigel, ECT = ectoderm, MES = mesoderm, END = endoderm. Scale bar = 100 µm. (**g**) Significantly more CAM area was invaded by *P1∆17* and *PL2L60* overexpressing OV-90 cells. Both mutants were more invasive than its respective wildtype form where only overexpression of *P1∆17* was significant. *n* = 4–18 per construct transfected and carried out in biological duplicates. Statistical tests Mann−Whitney U (non-normal distribution) or unpaired *t*-test (normal distribution) were used accordingly. * *p* < 0.05; *** *p* < 0.0005.

**Table 1 cancers-13-00004-t001:** Kaplan−Meier survival analysis of high grade serous ovarian cancer (HGSOC) patients according to piRNA pathway gene expression levels.

(a) Early Stage HGSOC Patients (FIGO Stage I and II)									
piRNA Pathway Genes	Progression Free Survival			Post Progression Survival			Overall Survival		
	HR	95% CI	*p*-Value	HR	95% CI	*p*-Value	HR	95% CI	*p*-Value
*PIWIL1*	0.66	0.31–1.4	0.27	**0.33**	**0.11–0.96**	**0.034**	**0.37**	**0.16–0.86**	**0.017**
*PIWIL2*	0.55	0.25–1.21	0.13	3.64	0.47–28.09	0.18	2.03	0.87–4.78	0.097
*PIWIL4*	2	0.74–5.4	0.16	0.17	0.02–1.48	0.07	0.66	0.21–2.1	0.48
*DDX4*	**0.45**	**0.2–1.01**	**0.046**	0.41	0.09–1.88	0.24	**3.51**	**1.02–12.09**	**0.035**
*HENMT1*	2.79	0.96–8.15	0.051	4.47	0.81–24.55	0.059	2.33	0.7–7.72	0.15
*MAEL*	**4.75**	**1.5–15.08**	**0.0037**	2.78	0.32–24.36	0.34	2.54	0.74–8.69	0.12
*PLD6*	0.42	0.14–1.29	0.12	0.3	0.04–2.23	0.21	3.4	0.74–15.61	0.094
*TDRD1*	**0.24**	**0.11–0.5**	**0.000042**	0.44	0.14–1.38	0.15	0.49	0.21–1.17	0.1
*TDRD9*	0.6	0.2–1.84	0.37	3.11	0.59–16.26	0.16	0.6	0.17–2.08	0.42
**(b) Late Stage HGSOC Patients (FIGO Stage III)**									
*PIWIL1*	0.91	0.76–1.08	0.27	**0.68**	**0.55–0.83**	**0.0002**	**0.7**	**0.58–0.85**	**0.00033**
*PIWIL2*	**1.32**	**1.09–1.61**	**0.0051**	**1.27**	**1–1.6**	**0.048**	0.88	0.73–1.06	0.18
*PIWIL4*	**1.35**	**1.06–1.72**	**0.015**	1.25	0.94–1.68	0.13	1.3	0.98–1.74	0.067
*DDX4*	0.91	0.76–1.09	0.3	1.17	0.96–1.43	0.11	0.88	0.72–1.07	0.19
*HENMT1*	1.17	0.9–1.52	0.23	1.2	0.87–1.66	0.27	1.28	0.93–1.74	0.13
*MAEL*	**1.27**	**1–1.62**	**0.048**	0.77	0.56–1.05	0.093	**0.72**	**0.53–0.99**	**0.041**
*PLD6*	1.19	0.94–1.51	0.15	1.18	0.88–1.59	0.27	**1.38**	**1.04–1.84**	**0.027**
*TDRD1*	0.89	0.74–1.08	0.24	0.84	0.67–1.05	0.14	0.83	0.69–1.01	0.061
*TDRD9*	**1.27**	**1–1.61**	**0.05**	0.82	0.6–1.11	0.2	1.18	0.89–1.56	0.24

Sample size of early stage HGSOC patients assessed for PFS (*n* = 84), PPFS (*n* = 32) and OS (*n* = 87); late stage HGSOC patients for PFS (*n* = 807), PPFS (*n* = 573) and OS (*n* = 836). HR > 1: low expression confers better outcome; HR < 1: high expression confers better outcome. HR = hazard ratio; 95% CI = 95% confidence interval; *p* < 0.05 = significant. HR, 95% CI and *p*-values in bold are significant.

## Data Availability

The data presented in this study are available in this article and Appendix A.

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
