# Peer review of "A Comprehensive Molecular and Clinical Analysis of the piRNA Pathway Genes in Ovarian Cancer"

_cancers, 2020, doi:10.3390/cancers13010004_

Round 1

Reviewer 1 Report

The manuscript submitted by Lee et al. describes the role of piRNA pathway genes in tumor development of HGSOC. This paper provides very interesting data but it still needs a considerable revision to be acceptable for the Cancers.

  1. Where is Figure 3?
  2. The authors presented in vitro motility and invasion after overexpressing several piRNA genes and their mutants using OV-90 and OVCAR-3 in Figure 5. In OV-90, only PIWIL2 mutant overexpression increased invasion compared to wildtype. Furthermore, only PIWIL1 mutant overexpression increased the motility compared to wildtype. This result lacks the consistency to describe the title of result 2.6. Could the authors show the same result using other HGSOC cell lines?
  3. Are the piRNA genes that the authors described the relevance with HGSOC specific to HGSOC? Did the authors examine other histotypes?
  4. It wasn’t clear to me the invasion of cancer cells to ECT in Figure 6c. The authors should show a better picture to clarify the invasion. 
  5. Line 280. I can't find the resulting figure described the piRNAs have relevance to chemoresistance.

Author Response

AU: We would like to thank the reviewer for the thoughtful comments. Below we provide information about how we revised the manuscript to address those comments.

1. Where is Figure 3?

AU: we apologise for this, the figure must have been deleted in the submission process as it is there in our pdf. We will check carefully when submitting the revised version to make sure all figures and tables are included. It can now be found on page 9.

2. The authors presented in vitro motility and invasion after overexpressing several piRNA genes and their mutants using OV-90 and OVCAR-3 in Figure 5. In OV-90, only PIWIL2 mutant overexpression increased invasion compared to wildtype. Furthermore, only PIWIL1 mutant overexpression increased the motility compared to wildtype. This result lacks the consistency to describe the title of result 2.6. Could the authors show the same result using other HGSOC cell lines?

AU: we have revised the title of 2.6 (page 10, lines 248/249) to better reflect our results (2.6. Cells overexpressing wildtype and mutant piRNA pathway genes have altered motility and invasion in vitro). PIWIL1 mutant significantly increased motility in OVCAR-3 and PIWIL2 mutant significantly increased invasion in OV-90. However, we do see similar trends for PIWIL2 mutant in the motility of OV-90 and PIWIL1 mutant in the invasion of OVCAR-3 and OV-90. Further testing on another HGSOC cell line is unfortunately beyond the scope of this project. We also think that further weight is added as increased invasion of PIWIL1 and PIWIL2 mutants was also reflected in the in vivo invasion assay.

3. Are the piRNA genes that the authors described the relevance with HGSOC specific to HGSOC? Did the authors examine other histotypes?

AU: we mainly focused the investigation on HGSOC and did a small part investigating low grade ovarian cancer (different histotype) as described in 2.2. we agree with the notion that there are limited other studies investigating piRNA pathway genes in epithelial ovarian cancer (please see discussion page 14, lines 417-420.)

4. It wasn’t clear to me the invasion of cancer cells to ECT in Figure 6c. The authors should show a better picture to clarify the invasion.

AU: we carefully went through our pictures and believe the one in the manuscript is the best suited to illustrate the invasion in this in vivo assay. We think that some of the confusion was caused by poor location of the asterisk and we have changed this which will help the reader to see the affected area of interest (page 12).

5. Line 280. I can't find the resulting figure described the piRNAs have relevance to chemoresistance.

AU: again we apologise for the loss of figure 3 in the submission process and this figure is retained in the revised submission on page 9.

We would like to thank the reviewer for the careful assessment of the ms and for spotting omissions and helping us improve the paper. We hope that we have addressed concerns in the revised manuscript.

Reviewer 2 Report

This paper by Lee E et al, presents expression screening of piRNA pathway genes in HGSOC tumors, benign serous cystadenoma and low grade OC tumors. The manuscript is straightforward, well written, and concise, and has clear results. Definitely deserves to be published and is a valuable contribution to the “cancers” journal. Some minor flaws need to be addressed before publication.

Minor points:

[1] “Introduction, Page 2/25, Lines 43-45”:

Epithelial ovarian cancer (EOC) is the main form of OC where a majority of malignant EOC consists of the very heterogeneous, high grade serous ovarian cancer (HGSOC [8].”.

Here, it should similarly reported that non-epithelial ovarian cancers account for about 10% of all ovarian cancers, and include malignancies of germ cell origin, sex cord-stromal cell origin, small-cell carcinomas and sarcomas.

Recommended reference: Boussios S, et al. Ovarian sex-cord stromal tumours and small cell tumours: Pathological, genetic and management aspects. Crit Rev Oncol Hematol. 2017;120:43-51.

[2]Discussion”, Page 12/25, Lines 285-288:

A number of studies reported that increased expression of piRNA pathway genes contributed to the metastatic progression of various cancers, including OC [55,58-61]. Limited information is available in OC and while knockdown of pathway genes has been linked to increased malignancy in vitro, overexpression analysis in OC surprisingly revealed decreased invasion [24,55-57]. ”.

Definitely, there is evidence for miRNA with demonstrated efficacy in decreasing the metastasis of tumor cells. Furthermore, please make a comment about the fact that they can also target epithelialmesenchymal transition (EMT) with potential clinical applications. Such miRNA and piRNA are potential candidates for translational medicine in ovarian cancer treatment.

Recommended reference: Palma Flores C, et al. MicroRNAs driving invasion and metastasis in ovarian cancer: Opportunities for translational medicine (Review). Int J Oncol. 2017 May;50(5):1461-1476.

[3] Discussion”, Page 12/25, Lines 311-314:

It is accepted that low grade and high grade OC are considered different diseases. Interestingly, this is also reflected in the piRNA pathway expression profile demonstrated by the significantly lower expression of PIWIL1, PIWIL3, PLD6 and TDRD1in high grade as compared to low grade. HENMT1, on the other hand, had elevated expression in high grade compared to low grade OC.”.

Furthermore, Iorio et al. used microarray analyses to compare serous, endometrioid, and clear cell tumors with normal ovarian tissues, and found that while some miRNAs were commonly up- or down-regulated among different subtypes of epithelial ovarian carcinomas, some miRNAs were dysregulated only in a specific subtype.

Recommended reference: Iorio MV, et al. MicroRNA signatures in human ovarian cancer. Cancer Res. 2007 Sep 15;67(18):8699-707.

[4]Conclusions”, Page 17/25, Lines 543-545:

Differential expression of these piRNA pathway genes in OC depicts a more complex association of their expression with various aspects of malignancy, patient survival, chemoresistance, gonadotropin treatment and invasion.”.

However, several obstacles limit their application in clinical practice. The different selection criteria for patients, collection methods and processing of biological samples may contribute to the different piRNA signatures obtained. Therefore, an optimal approach for piRNA detection should be followed, focusing on the variability in patients’ characteristics, experimental design, as well as the isolation and detection methodologies.

[5] A workflow diagram for the study would be of benefit for the readers.

Author Response

due to formatting issues we attach the response letter to this reviewer below. Thanks Frank Grutzner

Round 2

Reviewer 1 Report

The manuscript has been much improved and is in nice condition now.